# Illumina SBS Sequencing and DNBSEQ Perform Similarly for Single-Cell Transcriptomics

**DOI:** 10.3390/genes15111436

**Published:** 2024-11-06

**Authors:** Nadine Bestard-Cuche, David A. D. Munro, Meryam Beniazza, Josef Priller, Anna Williams, Andrea Corsinotti

**Affiliations:** 1Centre for Regenerative Medicine, Institute for Regeneration and Repair, MS Society Edinburgh Centre for MS Research, University of Edinburgh, Edinburgh EH16 4UU, UK; nbestard@ed.ac.uk (N.B.-C.);; 2UK Dementia Research Institute, University of Edinburgh, Edinburgh EH8 9AL, UK; 3Department of Psychiatry and Psychotherapy, School of Medicine and Health, Klinikum Rechts der Isar, Technical University Munich, and German Center for Mental Health (DZPG), 80333 Munich, Germany

**Keywords:** single-cell RNAseq, DNBSEQ sequencing, Illumina SBS sequencing

## Abstract

Background/Objectives: High-throughput single-cell RNA sequencing (scRNA-seq) workflows produce libraries that demand extensive sequencing. However, standard next-generation sequencing (NGS) methods remain expensive, contributing to the high running costs of single-cell experiments and often negatively affecting the sample numbers and statistical strength of such projects. In recent years, a plethora of new sequencing technologies have become available to researchers through several manufacturers, often providing lower-cost alternatives to standard NGS. Methods: In this study, we compared data generated from mouse scRNA-seq libraries sequenced with both standard Illumina sequencing by synthesis (Illumina SBS) and MGI’s DNA nanoball sequencing (DNBSEQ). Results: Our findings reveal similar overall performance using both technologies. DNBSEQ exhibited mildly superior sequence quality compared to Illumina SBS, as evidenced by higher Phred scores, lower read duplication rates and a greater number of genes mapping to the reference genome. Yet these improvements did not translate into meaningful differences in single-cell analysis parameters in our experiments, including detection of additional genes within cells, gene expression saturation levels and numbers of identified cells, with both technologies demonstrating equally robust performance in these aspects. The data produced by both sequencing platforms also produced comparable analytical outcomes for single-cell analysis. No significant difference in the annotation of cells into different cell types was observed and the same top genes were differentially expressed between populations and experimental conditions. Conclusions: Overall, our data demonstrate that alternative technologies can be applied to sequence scRNA-seq libraries, generating virtually indistinguishable results compared to standard methods, and providing cost-effective alternatives.

## 1. Introduction

Single-cell RNA sequencing (scRNA-seq) has revolutionized our ability to profile gene expression across thousands of individual cells, offering unprecedented insights into cellular heterogeneity and function. This technique holds immense value for numerous research and clinical applications [1,2]. While competition is slowly driving down scRNA-seq reagent prices, standard next-generation sequencing (NGS) methods for library sequencing remain expensive. These high running costs can lead researchers to use fewer samples for experiments, resulting in reduced statistical strength in single-cell experimental projects. The increasing demand for more cost-effective, high-quality short-read data, fuelled by emerging applications such as scRNA-seq and other workflows, has contributed to new NGS technologies being developed by new companies entering the sequencing market [3].

Illumina sequencing by synthesis (Illumina SBS) is the most established second-generation sequencing technology and has been the primary choice for most researchers due to its reliability and widespread adoption, with more than 90% of the world’s data being generated with its technology [4]. MGI’s DNA nanoball sequencing (DNBSEQ) platform is one of several alternative technologies seeking to offer competitive solutions. Both DNBSEQ and Illumina SBS utilize a biochemical process to amplify template DNA on flow-cell surfaces, enhancing fluorescent signals. However, they differ in the amplification method and the fluorescence used. With the Illumina SBS technique, the amplification is performed through bridge amplification, while with the DNBSEQ technology DNA fragments undergo rolling circle amplification, where each clonal copy of DNA is amplified from the template circularized DNA fragment, forming spherical “nanoballs” of amplified DNA. With Illumina SBS, the nucleotides are labelled with fluorescent dyes, while DNBSEQ uses nucleotides labelled with fluorescently labelled antibodies [3,5].

This study aims at comparing the performance of these two sequencing technologies specifically for single-cell transcriptomics. While previous studies have evaluated these platforms for bulk RNA-seq [6,7], the only existing reports for single-cell RNA-seq [8,9] have potential conflicts of interest, as they were funded by the manufacturers of the technologies. For the comparison we used data from *Csf1r*^ΔFIRE/ΔFIRE^ transgenic mice, as extensive transcriptomic changes have been reported in this mouse model [10]. *Csf1r*^ΔFIRE/ΔFIRE^ mice are designed to lack microglia, the main innate immune cell type in the central nervous system, and we showed that this deficiency altered the transcriptomic profiles of other brain cell types, especially oligodendrocytes, which specialize in producing myelin to insulate neuronal axons. These transgenic mice have a population of dysregulated oligodendrocytes that are not found in wild-type control mice, and these cells express *Serpina3n* and *C4b*, markers documented to be associated with demyelination, ageing and disease [11]. This clear difference between the datasets allowed us to achieve our aims and not only compare the technical sequencing parameters but also assess whether the sequencing method used influenced the biological outcomes.

The 10X Genomics libraries from *Csf1r*^ΔFIRE/ΔFIRE^ mice had been sequenced on Illumina Novaseq 6000 (Illumina, San Diego, CA, USA) as published in [12]. These same libraries were also converted into DNBSEQ-compatible libraries and were sequenced on an MGI DNBSEQ-T7 instrument (MGI, Shenzhen, China). Therefore, this study does not assess the library preparation differences but focuses solely on their sequencing.

We found that the sequencing quality (based on standard sequencing QC metrics) was slightly better for DNBSEQ than Illumina SBS, but this did not translate into meaningful differences in downstream biological observations, indicating equivalence between the two methods for single-cell transcriptomics studies.

## 2. Materials and Methods

### 2.1. Single-Cell Datasets and Cell Collection

To compare both NGS technologies, we used internally generated scRNA-seq libraries using 10X Genomics 3’ V3.1 reagents. The libraries were generated from the thalamus of the brain of *Csf1r*^ΔFIRE/ΔFIRE^ mice and matched controls, at an aged timepoint (17–19 months) [12].

The methods followed are as described previously in [12]. Briefly, brain tissue harvests and dissociations were performed at a consistent time of day for all eight mice, with processing of mice of different genotypes being mixed to reduce batch effects. Tissues were rapidly processed without perfusion, and thalami were dissected from both hemispheres and processed. We generated brain single-cell suspensions using the Adult Brain Dissociation Kit (130-107-677, Miltenyi Biotec, Surrey, UK). Cell sorting was conducted to exclude doublets, erythrocytes and debris. Post-sorting viability confirmation was performed, and single cells were processed using the Chromium Single Cell Platform (Pleasanton, CA, USA) (for details of each step, see [12]).

### 2.2. Sequencing

#### 2.2.1. Illumina SBS Sequencing

The library was quantified with Qubit and real-time PCR, while size distribution was assessed using an Agilent Technologies Bioanalyzer 2100 fragment analyzer (Agilent, Santa Clara, CA USA). Quantified libraries were pooled with other samples according to effective library concentration and data amount required. The library was sequenced on an Illumina NovaSeq 6000 instrument with an S4 flow cell as follows: 150 bp (Read 1), 2 × 10 bp (Indexes) and 150 bp (Read 2).

#### 2.2.2. DNBSEQ Sequencing

Illumina-compatible libraries generated using the 10X Genomics 3’ V3.1 workflow were converted for MGI sequencing using the PCR-based MGI Easy Universal Library Conversion kit (App-A) (MGI, Shenzhen, China) following the manufacturer’s instructions. The kit adds a 5’phosphate modification to the 10X library to enable circularization and subsequent DNA nanoball generation as part of the DNBSEQ sequencing process. Inputs of 50 ng were used, along with five cycles of PCR, to avoid the possibility of introducing errors through PCR. The libraries were sequenced on an MGI DNBSEQ-T7 instrument with an E1 flow cell. The sequencing consisted of 28 cycles for Read 1, 90 cycles for Read 2 and 10 cycles for the barcode indexes.

### 2.3. Bioinformatics

Illumina sequencing had been set to 150 bp reads due to the cycle configuration of an S4 flow cell, while for DNBSEQ, the 10X Genomics recommendations were followed [13]. To facilitate the comparison, all reads were cropped to 90 bp R2 and 28 bp R1 with Trimmomatic (v 0.39) [14]. To ensure that the differences in read depth would not influence the results, all samples were down-sampled to 300 M reads with the BBMAP reformat tool (v38.11) [15]. FastQC (v 0.11.9) provided general sequencing parameters, aggregated with MultiQC (v1.14) [16,17].

From the FastQC MultiQC reports, we extracted the Phred-Score from the ‘Per sequence quality scores’ plot raw data, and for each sample we obtained the mean Phred-Score by averaging all the reads per sample. The duplicated reads per sample were also obtained from the report, where they are defined as the percentage of the library that would be lost if all reads with exactly the same sequence were reduced to one single copy. This includes both technical duplicates (from PCR) and biological duplicates (due to the over-sequencing necessary to capture lowly expressed genes).

The alignment and initial cell calling were performed with CellRanger (v 7.0.0) using the default mm10 genome supplied by 10X Genomics (refdata-gexmm10-2020-A.tar.gz). We then combined the CellRanger reports with MultiQC and imported the text output files in R (v 4.2.1). From the CellRanger MultiQC reports we extracted the number of cells and reads; the mapping quality with the percentage of reads mapped to overall genome and to the intronic, exonic and antisense regions, and the sequencing saturation, defined as 1-n_unique_reads/n-reads).

Single cell RNA-seq analysis was performed with Bioconductor tools, namely Scater for QC [18], scran for normalization [19], variance estimation and dimensional reduction, and SingleR [20] to annotate the cells. The thresholds for the strict quality control were a minimum of 3000 UMI counts and 750 genes detected and a maximum of 6% mitochondrial genes. Differential expression analysis between *Csf1r*^ΔFIRE/ΔFIRE^ and *Csf1r*^WT/WT^ samples was performed on the subsetted oligodendrocytes with MAST (v 1.24.1) [21]. To plot gene metrics divided by technology (Figures 2C and 4C), the average values between the two technologies were used for the genes detected in both.

### 2.4. Statistics

The difference between the average Phred-Score and, the other paired values, such as the percentage of duplicated reads per sample, the percentage of reads mapped to genome or the sequencing saturation, were tested with a two-tailed paired *t*-test (*n* = 8, df = 7).

The correlation between technologies in the cell numbers per gene and UMIs detected per cell were explored by fitting a linear model, reporting the coefficient of determination (R^2^) on the plots.

To test if there was a significant difference in the distribution of cells among the cell types between the two sequencing methods, we used a Chi-squared test of independence (df = 10). We also used Chi-squared of independence to test if there was a difference in the proportion of cells excluded during quality control (df = 1).

## 3. Results

### 3.1. DNBSEQ Shows Higher Sequencing Quality But No Improvement of Single-Cell RNA-Seq Metrics

Due to the configuration requirements of an S4 flow cell, the Illumina SBS sequencing reads were longer (150 bp) than the ones generated with DNBSEQ (28–90 bp). We trimmed all reads to the same length to make them comparable. The reads were cropped to match the 10X Genomics recommendations [13]. This is 90 bp for Read 2, that contains the RNAseq insert, and 28 bp for Read 1 that contains the cell barcode and the unique molecular identifier (UMI). All libraries also greatly differed in sequencing depth, ranging from 300 M to 400 M for both technologies. Therefore, to ensure an even comparison, all libraries were randomly down-sampled to 300 M reads each.

To evaluate read quality, we assessed Phred-scaled quality scores, which represent the confidence in base call assignments made by the sequencer [22]. A lower Phred score (<Q20) indicates a lower base call accuracy, while Phred scores over 30 (>Q30) are considered a benchmark for quality in next-generation sequencing with a probability over 99.9% of bases being accurately called without ambiguities [23]. Both technologies achieved a Phred quality score above Q30 for over 95% of bases. The average Phred quality score for the samples sequenced with DNBSEQ was 35.76 ± 0.09, while the average for Illumina SBS was 35.50 ± 0.05. Although the two values were very similar, a paired *t*-test with the average for each sample (*n* = 8) showed a significant difference between the means (*p*-value = 4.29 × 10^−6^, df = 7) (Figure 1A).

Another commonly used sequence quality indicator is the percentage of duplicated reads, defined as the percentage of the library that would be lost if all reads with exactly the same sequence were reduced to one single copy. There are two potential types of duplicates, technical and biological. The technical duplicates are due to the PCR amplification steps used during library preparation. The other source of duplication is biological duplicates, where different copies of the same gene with identical sequence are randomly selected. Both kinds of duplicates are expected in scRNA-seq, as cDNA PCR amplification and deep sequencing are necessary to allow detection of lowly expressed transcripts; consequently, highly expressed transcripts are commonly over-sequenced [16]. This potentially creates a large set of duplicates, as seen in both DNBSEQ and Illumina SBS techniques where the duplicates were over 60% for all samples, with Illumina SBS having a slightly greater, although statistically significant, percentage of duplicates (Figure 1B).

To assess whether the differences in read quality (Q30) and duplicates affected the read alignment to the reference genome, we proceeded with the alignment using CellRanger, from 10X Genomics [24]. The most important metric to assess alignment quality is the mapping rate, which is defined as a proportion of reads mapped uniquely to the reference genome, out of all input reads. For high-quality libraries, the mapping rate should exceed 90%, while low mapping rates (<50%) generally indicate technical problems with sample quality, during library preparation/sequencing, or data processing [25]. We can further divide the mapping by whether reads map to areas that encode for exons or to intronic/antisense regions of the transcriptome. Most reads come from exonic regions that code for proteins and are the main focus of most scRNAseq analysis. A small proportion of reads fall into intronic regions from unspliced reads, or from antisense regions, from RNAs that have a regulatory function.

Following these criteria, we assessed the proportion of uniquely mapped reads for each sample. Both sets of samples had a read mapping score over 90%, indicating a high-quality mapping, with an average of 97 ± 0.48% for reads generated with DNBSEQ and 94.9 ± 0.31% for the Illumina SBS (Figure 1C), with a small but significant difference (*p*-value = 0.00149, *n* = 8). Dividing the mapping by genomic areas revealed that the additional percentage of reads from the DNBSEQ libraries mapping to the genome were mostly from antisense and intronic regions. The exonic mapping did not significantly differ between the two sequencing technologies (*p*-value = 0.283, *n* = 8), while the reads mapped to intronic and antisense regions were increased in the DNBSEQ libraries (*p*-value = 0.000181 and *p*-value = 0.00159, respectively, *n* = 8) (Figure 1D,E and Appendix A).

On top of the alignment to the reference genome, CellRanger performs an alignment to the transcriptome, barcode processing and cell calling. This yields additional metrics that can give an insight into the quality of the sequencing technologies specifically for scRNA-seq. Despite a significant difference in the percentage of mapped reads between the two technologies, this did not affect the single-cell-specific quality metrics. The sequencing saturation, which denotes the point of diminishing returns in discovering new genetic features despite increased sequencing depth, was comparable for both technologies (Figure 1F). The average number of cells detected, as well as the numbers of reads and genes per cell were also not significantly different between technologies (Appendix A).

### 3.2. Different Sequencing Technologies Yield Comparable Gene Expression Data

We next compared the cells and genes detected in scRNA-seq experiments using the two sequencing technologies. We observed that most cells and genes detected were overlapping between the samples sequenced with Illumina SBS or DNBSEQ technology (Figure 2A).

We then performed a stricter quality control and compared the remaining cells and genes that did not overlap post quality control. For cells, the quality control consisted of removing cells with low UMI counts (Figure 2B), low genes detected and high mitochondrial genes. These values normally indicate low-quality cells (e.g., dead cells). For genes, the quality control consisted of filtering for genes that were not expressed in at least ten cells (Figure 2C). After stricter quality control was carried out using these parameters, only three cells and two genes (marked in grey in the “only in DNBSEQ” and “only in Illumina SBS” groups) were not detected in both technologies (Figure 2B,C).

We next measured the correlation between sequencing methods from the cells and genes detected with both DNBSEQ and Illumina SBS sequencing technologies. The number of genes per cell had a correlation with an R^2^ of 0.9964 between sequencing methods (Figure 2D), and the average gene expression for each gene had a correlation of R^2^ 0.9958 (Figure 2E).

### 3.3. Similar Cell Populations Were Identified with Both Technologies Using Reference-Based Algorithms as Well as Marker Gene Expression

To further assess the concordance between the two datasets, we merged the data and conducted a comprehensive joint analysis. The most broadly applied method to visualize scRNA-seq is laying all cells into a reduced dimensional reduction space such as a T-SNE plot. The goal of a dimensional reduction for visualization is to place cells with a similar gene expression together and different cells further apart on the 2D plane. The dimensional reduction surprisingly revealed no batch effects attributable to the sequencing technology employed. With no integration, all cells were already overlayed in the T-SNE plot (Figure 3A), highlighting a high level of agreement between the two datasets and affirming their similarity.

Discerning individual cell types within a diverse population constitutes a pivotal initial stage in the analysis of scRNA-seq data as it lays the groundwork to further explore the system under investigation. Cell-type identification is typically accomplished using two main methods: reference-based scoring or manual annotation. For reference-based scoring, we used the reference dataset [26] and transferred the labels with singleR to our merged dataset containing the data from both technologies. The obtained labels are represented by color on the T-SNE plot in Figure 3B. We tested if there was a difference in the cell distribution among the different assigned cell types between the two sequencing technologies with a Chi-squared test of independence, and the analysis did not yield any significant difference in the labelling results (df = 10) (Figure 3C,D). For manual annotation, known markers are used to annotate cells. We used gene expression for markers of the four most abundant cell types: *Pdgfra* for oligodendrocyte progenitor cells (OPCs), *Mbp* for oligodendrocytes, *P2ry12* for microglia and *Cldn5* for endothelial cells. These markers showed the same expression distribution across cell types regardless of the sequencing method used (Figure 3E).

### 3.4. The Sequencing Method Used Minimally Affects Meaningful Biological Comparisons

Oligodendrocytes are the most affected cell type in the *Csf1r*^ΔFIRE/ΔFIRE^ genotype, with a new population of dysregulated oligodendrocytes appearing only in the depleted mice (Figure 4A), characterized by increased expression of the Serpina3n and C4b transcripts [26]. To evaluate if the same biological observation could be recapitulated independently of the sequencing technology used, we investigated whether the dysregulated gene expression of oligodendrocytes in the *Csf1r*^ΔFIRE/ΔFIRE^ mice [12] was similarly detectable in both datasets.

Differential expression analysis between control and *Csf1r*^ΔFIRE/ΔFIRE^ oligodendrocytes yielded 156 differentially expressed genes with DNBSEQ sequencing and 186 differentially expressed genes with Illumina SBS sequencing. Out of those, 152 genes were identified with both technologies (Figure 4B).

To better understand the differences between the two results, we compared the magnitude of the changes from the genes obtained with one or the two technologies. The genes only detected with one of the analyses had a smaller change (average abs (log2FC) = 0.26) between *Csf1r*^ΔFIRE/ΔFIRE^ and control than the genes obtained with both (average abs (log2FC) = 0.46). None of the genes only detected with one technology had a change greater than 0.3 abs (log2FC) (Figure 4C). When analyzing differential expression, results are often sorted by log2FC or *p*-value and the genes with maximum change are then explored further. When doing this, the top 10 hits in both classifications were identical between the analysis performed with each technology (Appendix A) and included our known marker genes *Serpina3n* and *C4b* [26] (Figure 4D,E).

## 4. Discussion

In this study, we compared MGI’s DNBSEQ technology with the broadly used bridge amplification sequencing technology from Illumina for scRNA-seq analysis to help researchers make an informed choice when selecting emerging sequencing alternatives.

Overall, both DNBSEQ and Illumina SBS technologies gave values of Phred-score and mapping to the genome associated with high-quality libraries [23,25]. DNBSEQ-converted libraries showed lower values of duplicated reads, possibly due to the linear DNA amplification (rolling circle DNA synthesis) method used by DNBSEQ compared to PCR amplification used by the Illumina SBS technology (Figure 1). The number of cells recovered was not significantly different between both technologies. Moreover, the cells only recovered in the DNBSEQ or Illumina SBS libraries were low-quality cells, which are likely just below or above the lenient CellRanger thresholds [27]. Most importantly, the cells and genes kept and detected with both technologies were highly comparable, with R2 gene expression and cell correlations exceeding 0.995 (Figure 2).

This similarity was emphasized when we merged the two datasets with no integration and observed no batch effects between the two technologies. Automated cell annotation using a reference dataset also showed no significant difference in assigned cell types between the two sequencing technologies (Figure 3). Finally, we performed a differential gene expression between wild-type and *Csf1r*^ΔFIRE/ΔFIRE^ oligodendrocytes and found that most differences were detected in both datasets, including our selected genes that we previously validated, and those that varied between the techniques had lower-fold change expression differences (Figure 4). Therefore, although our comparison showed statistically significant differences in sequencing parameters, the magnitude of this effect in the biological context is not likely of substantial relevance. Overall, we believe that these two sequencing technologies perform equivalently when applied to sequencing of single-cell RNA-seq datasets.

Our findings are consistent with previous studies comparing DNBSEQ and Illumina technologies for single-cell RNAseq and other transcriptomics. DNBSEQ has shown either equivalent or slightly improved read quality over Illumina platforms [6,8,28]. Others have studied the difference between these two technologies in bulk RNA-seq and have shown a high concordance in global gene expression [6], concordance in the dimensional space reduction [6,28] and equivalent differentially expressed genes [7,28]. Previous comparisons in these sequencing methods for scRNA-seq show equivalent accuracy and sensitivity [9]; this translates into similar metrics like cell and UMI detection as well as population identification [8]. The impact of the differences between these technologies will clearly depend on the question being answered; for example, DNBSEQ may detect more SNPs or INDELs in whole-genome variant calling [29], but when using SNPs to assign single-cells to the correct donor, there was no difference between the two platforms [8].

Manufacturers continue to introduce new sequencing alternatives. Here, we compared MGI’s DNBSEQ with the standard SBS chemistry from Illumina. MGI has already released an improved kit to convert the Chromium 10X libraries to DNBSEQ-compatible libraries, and Illumina is releasing its SBS-XLEAP technology. The SBS-XLEAP sequencing chemistry works on the new X and Xplus Illumina instruments and is sold as a cheaper alternative than the classic SBS chemistry, while providing higher-quality reads. In addition, competitors like PacBio, Element Biosciences and Ultima are also entering the short-read sequencing market [3,30]. Therefore, more studies on the state-of-the-art reagents for both technologies compared here, as well with other competitors, would be useful. However, for standard scRNA-seq applications, the read quality improvements from these technologies are likely to have less impact than the cost reductions driven by increased competition.

## Figures and Tables

**Figure 1 genes-15-01436-f001:**
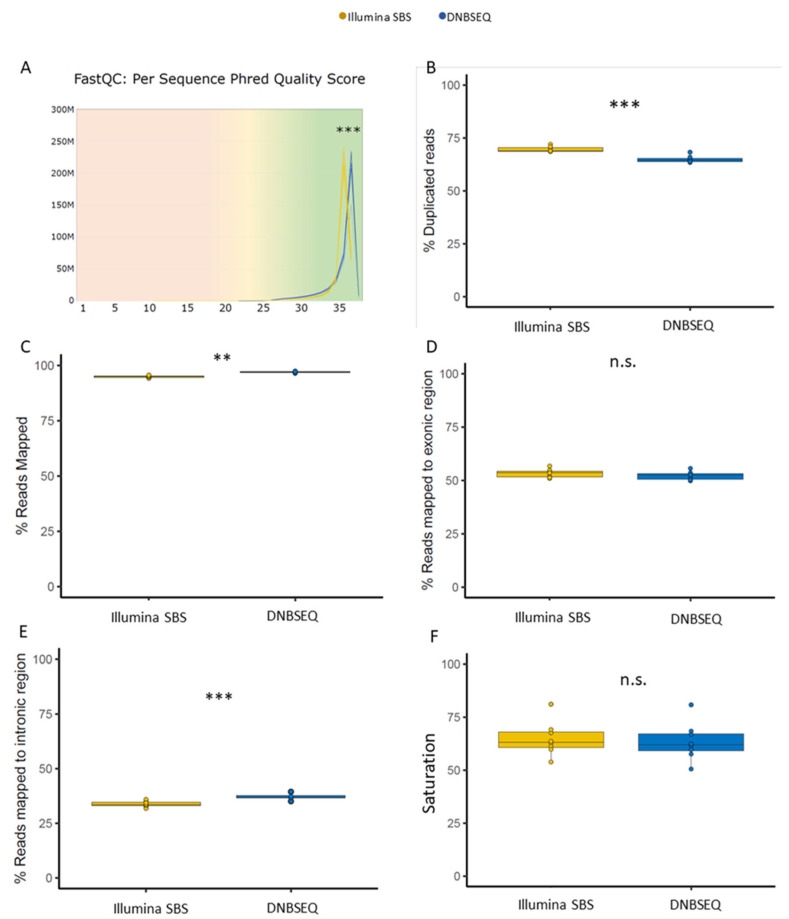
DNBSEQ shows higher sequencing quality but no improvement of single-cell RNA-seq metrics. (**A**) Read count distribution against Phred scores, calculated based on the average per-base Phred score for each Read 2. Each line represents a sample, with DNBSEQ-sequenced samples depicted in blue and samples sequenced with Illumina SBS in yellow (*n* = 8; Illumina SBS mean ± SD = 35.50 ± 0.05, DNBSEQ mean ± SD = 35.76 ± 0.09; *t*-test *p*-value 4.29 × 10^−6^). (**B**) Box plot representation of the percentage of duplicated reads in each sample (*n* = 8; Illumina SBS mean ± SD = 69.6 ± 1.3, DNBSEQ mean ± SD = 64.9 ± 1.6; *t*-test *p*-value = 1.78 × 10^−5^). (**C**) Box plot representation of the percentage of reads mapped to the mouse reference genome. (*n* = 8, Illumina SBS mean ± SD = 94.9 ± 0.48, DNBSEQ mean ± SD = 97.0 ± 0.31; *p*-value = 0.00149). (**D**) Box plot with the percentage of reads confidently mapped to exonic regions (*n* = 8; Illumina SBS mean ± SD = 53.3 ± 2.0, DNBSEQ mean ± SD = 52.2 ± 2.0; *t*-test *p*-value = 0.283). (**E**) Box plot with the percentage of reads confidently mapped to intronic regions (*n* = 8; Illumina SBS mean ± SD = 33.7 ± 1.3, DNBSEQ mean ± SD = 37.1 ± 1.4; *t*-test *p*-value = 0.000181). (**F**) Box plot representation of sequencing saturation, a measure of the fraction of reads originating from an already-observed UMI (*n* = 8, DNBSEQ mean ± SD = 63.5 ± 8.9, Illumina SBS mean ± SD = 64.9 ± 8.1; *t*-test *p*-value = 0.751). n.s. *p*-value > 0.05; ** *p*-value < 0.01, *** *p*-value < 0.001.

**Figure 2 genes-15-01436-f002:**
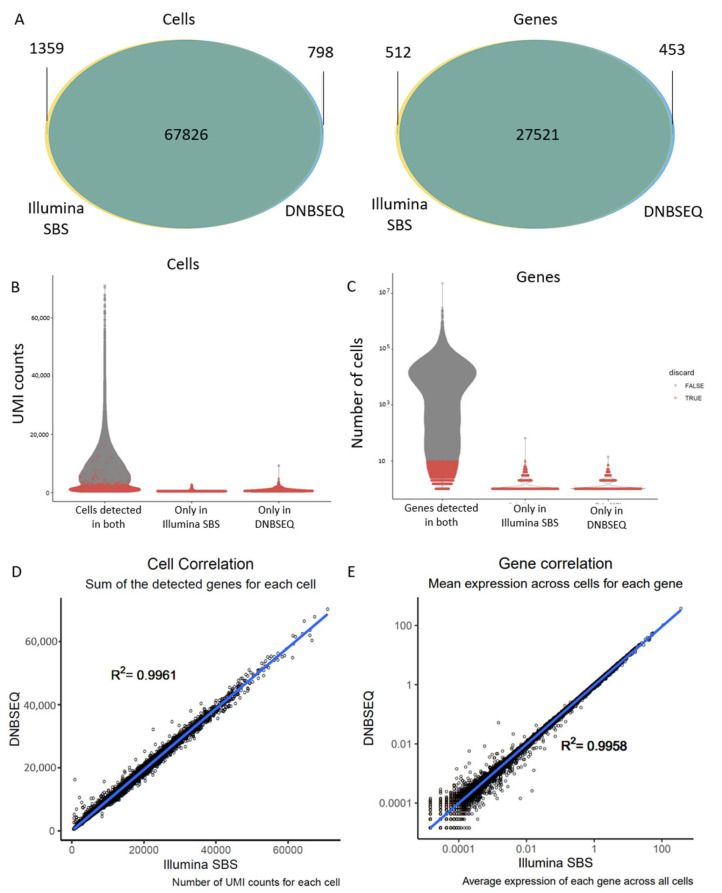
Different sequencing technologies yield comparable gene expression results. (**A**) Area-proportional Venn diagrams showing overlapping genes and cells detected with each technology before the strict quality control filtering. DNBSEQ is depicted in blue and Illumina SBS in yellow. (**B**) Violin plots showing the UMI count per cell. On the *x*-axis, cells are separated according to being detected with both technologies, or only one of the technologies. In red are the cells that were discarded after the stricter quality control. (**C**) Violin plot with the distribution of the number of cells where each gene is detected. On the *x*-axis, genes are separated according to being detected with both technologies or only one of the technologies. In red are the genes that were discarded after the stricter quality control. (**D**) Cell correlation analysis between both technologies, measured comparing the number of detected genes for each cell. (**E**) Gene correlation analysis between both technologies, measured comparing the average expression for each gene across cells.

**Figure 3 genes-15-01436-f003:**
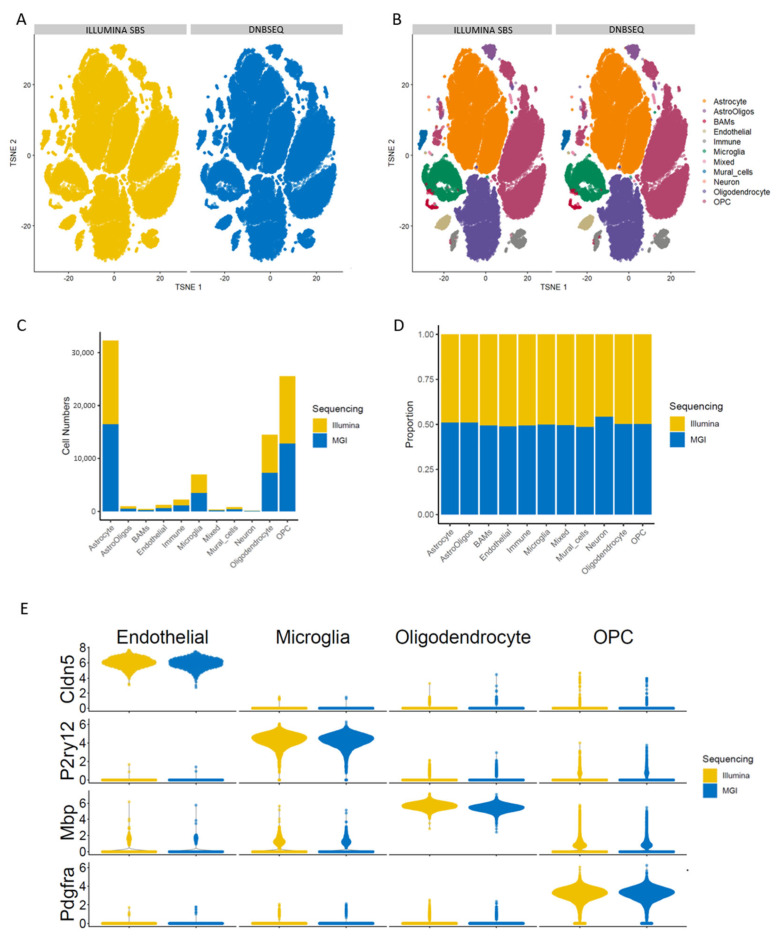
Similar cell populations were identified with both technologies using reference-based algorithms as well as marker gene expression. (**A**) T−SNE dimensional reduction computed with the data coming from both technologies. The cells are colored and split in the panels according to the technology used for sequencing. (**B**) T−SNE dimensional reduction colored with the assignment to the different brain cell types by transfer label. (**C**) Barplot with the number of cells assigned to each cell type from each technology. (**D**) Barplot with the proportion of cells from each technology assigned to each cell type. (**E**) Violin plot showing the gene expression distribution of biomarkers for the four most abundant cell types in this dataset as an illustration of similarity.

**Figure 4 genes-15-01436-f004:**
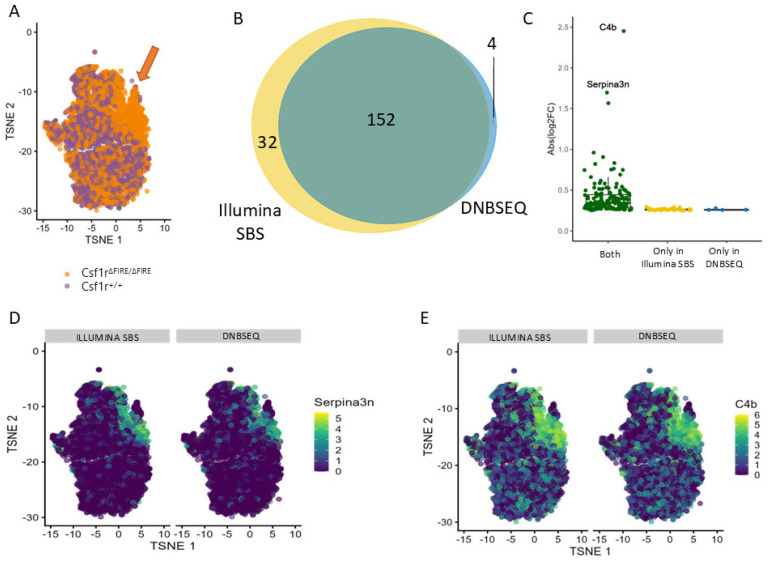
The sequencing method used minimally affects meaningful biological comparisons. (**A**) T−SNE plot with the subsetted oligodendrocytes showing the population unique to the *Csf1r*^ΔFIRE/ΔFIRE^ compared to its control (indicated by the arrow). (**B**) Venn diagram showing the overlap of results obtained with the DNBSEQ (blue) and the Illumina SBS (yellow) libraries for the differentially expressed genes between *Csf1r*^ΔFIRE/ΔFIRE^ and the control calculated with MAST. (**C**) Box plot with a comparison of the absolute log2FC expression obtained from the genes that were detected in both analyses, only in the analysis with the Illumina SBS technology and only obtained in the analysis with the DNBSEQ technology. (**D**) T−SNE with superimposed expression of *Serpina3n* compared between both technologies. (**E**) T−SNE with superimposed expression of *C4b* compared between both technologies.

## Data Availability

The original data presented in the study are openly available in GEO at https://www.ncbi.nlm.nih.gov/geo/query/acc.cgi?acc=GSE267545 (accessed on 31 October 2024) with accession number GSE267545 for the Illumina SBS sequencing. The DNBSEQ sequencing has been uploaded to GEO and the accession number will be made available to reviewers on request. The code used is available in GitHub under the repository https://github.com/SingleCellCRM/MGIvsIllumina_scRNAseq/ (accessed on 31 October 2024).

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
