# Peer review of "Illumina SBS Sequencing and DNBSEQ Perform Similarly for Single-Cell Transcriptomics"

_genes, 2024, doi:10.3390/genes15111436_

Round 1
Reviewer 1 Report
Comments and Suggestions for Authors
Bestard-Cuche et al. present a well-written, relevant manuscript. I found it easily understandable, but have some suggestions/comments for the authors:
1. Particularly, I like the aim to be presented by the end of the Introduction. The authors could first explain the transgenic model and then present the aim of the study.
2. When I first looked at the Materials and Methods section I found it short for a method comparison article. However, many of the aspects that should be in this section are in other parts of the manuscript (i.e., in the last paragraph of the Introduction section, the paragraph from line 181 in the Results). I suggest moving the information so the readers can understand all the quality controls performed by reading the Materials and Methods.
3. Please cite the reference genome/transcriptome used.
4. Since the authors mentioned bulk RNA-seq in their discussion, I suggest they add one or two sentences about the deconvolution methods and how they have helped evaluate the concordance of gene expression regardless of the technology used.
Author Response
Thank you very much for reviewing our article and the constructive comments. We have replied in full below:
- Particularly, I like the aim to be presented by the end of the Introduction. The authors could first explain the transgenic model and then present the aim of the study.
We have moved the aims presented before the transgenic model explanation to the end of the paragraph as suggested.
Sentence added at the end of the paragraph:
"This clear difference between the datasets allowed us to achieve our aims and not only compare the technical sequencing parameters, but also assess whether the sequencing method used influenced the biological outcomes. "
- When I first looked at the Materials and Methods section I found it short for a method comparison article. However, many of the aspects that should be in this section are in other parts of the manuscript (i.e., in the last paragraph of the Introduction section, the paragraph from line 181 in the Results). I suggest moving the information so the readers can understand all the quality controls performed by reading the Materials and Methods.
We moved part of the last paragraph of the introduction section to the methods.
"To compare both NGS technologies, we used internally generated scRNA-seq libraries using 10X Genomics 3’ V3.1 reagents. The libraries were generated from the thalamus of the brain of Csf1rΔFIRE /ΔFIRE mice, and matched controls, at an aged timepoint (17-19 months)12."
We added in the methods a clarification about duplicated reads (explained on former line 181 from the results) and we also added further explanation on the other metrics extracted from the MulitQC reports to the section.
"From the FastQC MultiQC reports we extracted the Phred-Score, from the `Per sequence quality scores` plot raw data, and for each sample we obtained the mean Phred-Score by averaging all the reads per sample. The duplicated reads per sample were also obtained from the report, where they are defined as the percentage of the library that would be lost if all reads with exactly the same sequence were reduced to one single copy. This includes both technical (PCR) and biological (due to inherent to the technology over-sequencing) reads."
"The alignment and initial cell calling were performed with Cellranger (v 7.0.0) using the default mm10 genome supplied by 10x Genomics (https://cf.10xgenomics.com/supp/cell-exp/refdata-gex mm10-2020-A.tar.gz). We then combined the cellranger reports with MultiQC and imported the text output files in R (v 4.2.1). From the cellranger MultiQC reports we extracted the number of cells and reads; the mapping quality with the percentage of reads mapped to overall genome and to the intronic, exonic and antisense regions and the sequencing saturation, defined as 1- n_unique_reads/n-reads)."
- Please cite the reference genome/transcriptome used. This has been added.
4. Since the authors mentioned bulk RNA-seq in their discussion, I suggest they add one or two sentences about the deconvolution methods and how they have helped evaluate the concordance of gene expression regardless of the technology used.
The bulk RNAseq in the discussion is not deconvoluted to single cell level. The bulk RNAseq was added as an example of another transcriptomics application where these technologies had been used and compared. We clarified this in the manuscript.
"Our findings are consistent with previous studies comparing DNBSEQ and Illumina technologies for single-cell RNAseq and other transcriptomics. DNBSEQ has shown either equivalent or slightly improved read quality over Illumina platforms[6,8,29]. Others have studied the difference between these two technologies in bulk RNA-seq and have shown a high concordance in global gene" expression[6],
Reviewer 2 Report
Comments and Suggestions for Authors
The manuscript by Nadine Bestard-Cuche and colleagues describes a comparison of Illumina SBS sequencing with the more recently-introduced MGI DNA Nanoball Sequencing platform to determine whether the two platforms yield similar results for a single-cell transcriptomics study. The experimental system used in the study is a transgenic mouse model for which previous transcriptomic studies have been conducted.
Overall, this study is straightforward and its design and conclusions appear to be sound. The experimental methods are appropriate for the study's primary objectives and questions, and the results support the conclusions.
Specific comments are presented below, in order of line number.
Lines 63-64: The introduction states that the only previous evaluations of the two technologies for sc-RNASeq have potential conflicts of interest. If this is the case, then the nature of those conflicts should be explained.
L. 90: "hemisphere" should be "hemispheres"
L. 116-117: insert a comma after "results"
L. 140-142: explain which Chi-squared test (e.g., test of independence or goodness-of-fit test) was used.
In the Results section, statistical test results should report the number of degrees of freedom.
Also in the Results section, there are several places (L. 243-244, L. 282-283, L. 314-315) where an interpretation of the results is given. These statements seem unnecessary as the same conclusion is made in the Discussions, which in my opinion is a more appropriate place for such interpretations.
L. 246: recommend changing "gene expression" to "gene expression results".
L. 318-320. In the caption for Figure 4, part A, explain what the arrow is pointing to.
L. 351: remove comma after "context"
L. 359: replace comma after "sensitivity" with a semicolon
Author Response
Thank you very much for reviewing our article. We have replied in full below:
Comment 1: Lines 63-64: The introduction states that the only previous evaluations of the two technologies for sc-RNASeq have potential conflicts of interest. If this is the case, then the nature of those conflicts should be explained.
Response 1: We have added the explanation.
Comment 2: 90: "hemisphere" should be "hemispheres"
Response 2: Changed - thank you
Comment 3: 116-117: insert a comma after "results"
Response3: Changed - thank you
Comment 4: 140-142: explain which Chi-squared test (e.g., test of independence or goodness-of-fit test) was used.
Response 4 The clarification was added to methods and results – thank you
Comment 4 In the Results section, statistical test results could report the number of degrees of freedom.
Response 4 We added the degrees of freedom to the statistics section of the methods for all statistical tests and to the results too for the chi-squared test
Comment 5 Also in the Results section, there are several places (L. 243-244, L. 282-283, L. 314-315) where an interpretation of the results is given. These statements seem unnecessary as the same conclusion is made in the Discussions, which in my opinion is a more appropriate place for such interpretations.
Response 5: We have removed these summary statements.
Comment 6: 246: recommend changing "gene expression" to "gene expression results".
Response 6 Changed - thank you
Comment 7: 318-320. In the caption for Figure 4, part A, explain what the arrow is pointing to.
Response 7 We have added explanation to the legend.
Comment 8: 351: remove comma after "context"
Response 8 Changed - thank you
Comment 9: 359: replace comma after "sensitivity" with a semicolon
Response 9 Changed - thank you